# Accompanying Hemoglobin Polymerization in Red Blood Cells in Patients with Sickle Cell Disease Using Fluorescence Lifetime Imaging

**DOI:** 10.3390/ijms252212290

**Published:** 2024-11-15

**Authors:** Fernanda Aparecida Borges da Silva, João Batista Florindo, Amilcar Castro de Mattos, Fernando Ferreira Costa, Irene Lorand-Metze, Konradin Metze

**Affiliations:** 1Departments of Pathology and Internal Medicine, Faculty of Medical Sciences, State University of Campinas, Campinas 13083-887, Brazil; fernandaborgesilva@gmail.com (F.A.B.d.S.); castrom@unicamp.br (A.C.d.M.); 2National Institute of Science and Technology on Photonics Applied to Cell Biology (INFABIC), State University of Campinas, Campinas 13083-970, Brazil; 3Institute of Mathematics, Statistics, and Scientific Computing, State University of Campinas, Campinas 13083-888, Brazil; florindo@unicamp.br; 4Laboratory of Pathology, Pontifical Catholic University of Campinas PUCC, Campinas 13060-904, Brazil; 5Department of Internal Medicine, Faculty of Medical Sciences, State University of Campinas, Campinas 13083-859, Brazil; ferreira@unicamp.br (F.F.C.); ilmetze@unicamp.br (I.L.-M.)

**Keywords:** fluorescence lifetime imaging, auto-fluorescence, hemoglobin, polymerization, machine learning

## Abstract

In recent studies, it has been shown that fluorescence lifetime imaging (FLIM) may reveal intracellular structural details in unstained cytological preparations that are not revealed by standard staining procedures. The aim of our investigation was to examine whether FLIM images could reveal areas suggestive of polymerization in red blood cells (RBCs) of sickle cell disease (SCD) patients. We examined label-free blood films using auto-fluorescence FLIM images of 45 SCD patients and compared the results with those of 27 control persons without hematological disease. All control RBCs revealed homogeneous cytoplasm without any foci. Rounded non-sickled RBCs in SCD showed between zero and three small intensively fluorescent dots with higher lifetime values. In sickled RBCs, we found additionally larger irregularly shaped intensively fluorescent areas with increased FLIM values. These areas were interpreted as equivalent to polymerized hemoglobin. The rounded, non-sickled RBCs of SCD patients with homogeneous cytoplasm were not different from those of the erythrocytes of control patients in light microscopy. Yet, variables from the local binary pattern-transformed matrix of the FLIM values per pixel showed significant differences between non-sickled RBCs and those of control cells. In a linear discriminant analysis, using local binary pattern-transformed texture features (mean and entropy) of the erythrocyte cytoplasm of normal appearing cells, the final model could distinguish between SCD patients and control persons with an accuracy of 84.7% of the patients. When the classification was based on the examination of a single rounded erythrocyte, an accuracy of 68.5% was achieved. Employing the Linear Discriminant Analysis classifier method for machine learning, the accuracy was 68.1%. We believe that our study shows that FLIM is able to disclose the topography of the intracellular polymerization process of hemoglobin in sickle cell disease and that the images are compatible with the theory of the two-step nucleation. Furthermore, we think that the presented technique may be an interesting tool for the investigation of therapeutic inhibition of polymerization.

## 1. Introduction

Sickle cell disease (SCD) is a group of inherited diseases defined by mutations in the gene encoding the beta globin chain of hemoglobin, which is the oxygen carrier molecule of red blood cells (RBCs) [1,2]. This altered hemoglobin has a lower affinity for O_2_ and polymerizes in the deoxygenated form, leading to a deformation of RBCs (sickling). These cells have impaired rheology and a premature hemolysis. The most common disease is sickle cell anemia (homozygous SS), but hemoglobin S can also be co-inherited with hemoglobin C (hemoglobinopathy SC) or thalassemia (Sβ-thalassemia or Sα-thalassemia). All these disorders have a similar clinical spectrum of vaso-occlusive painful crises and multiple organ damages. 

SCD is prevalent in several regions of the world, such as Africa, the USA, and the Middle East, and is now increasing in Europe. In Brazil, an incidence between 0.15% and 10.9% of the βS allele has been reported. Therefore, an early diagnosis, especially in children and in primary healthcare as well as in blood banking, is important.

The key pathophysiological process in SCD is the hemoglobin polymerization, which has been analyzed by different sophisticated laboratory methods [3,4,5,6,7,8,9].

The majority of the studies have been made in vitro, and thus the topographical context of the molecules was lost. Furthermore, artifacts may be introduced. In this context, methods based on molecule fluorescence are interesting because they permit us to analyze the polymerization process without alteration of the cell topology. In a previous work, we described the FLIM characteristics of normal RBCs in label-free routine cytological films of normal bone marrow [10].

Recently, Garcia and Blum demonstrated that fluorescence lifetime imaging (FLIM) is able to examine the phenomenon of polymerization of molecules in a real-time manner [11]. The authors have also shown a direct correlation between polymer fluorescence lifetime and its molecular weight and could morphologically follow the temporal changes in the polymer architecture over time in a liquid medium. Therefore, the application of this method for the study of hemoglobin S polymerization could be interesting. 

FLIM microscopy permits the creation of virtual images showing sub-cellular structures not seen in standard preparations [11,12,13,14,15,16,17]. The image contrast is built by differences between the fluorescence lifetimes, which are captured per pixel of the image. When a molecule is hit by a laser pulse, there is a time delay, called fluorescence lifetime before a fluorescence photon will be emitted (fluorescence lifetime). Several molecules are auto-fluorescent, such as FAD, NADH, NADPH, hemoglobin, porphyrins, fatty acids and lipopigments, tryptophan, and retinol [11,13,14,15,16,17]. This fluorescence is captured by a device that registers the lifetime of individual photons. Mean lifetimes/pixel are calculated and transformed into pseudo-colors, thus creating virtual microscopic images [11]. The lifetime is not only dependent on the chemical composition of the molecule but also on its environment (pH, pO_2_) and conformational changes, including polymerization [11,13,14,15,16,17]. 

Taking into account that FLIM features may describe the attributes of polymerized proteins in routine laboratory cytological and histological material, the aims of the present study were:

To examine if hemoglobin polymerization is detectable by FLIM in routinely prepared, label-free peripheral blood smears of patients with SCD and to compare these images with those obtained by confocal autofluorescence. 

To extract texture matrices from the FLIM images (fluorescence lifetime data per pixel) and compare the values obtained in SCD and normal controls.

To quantify neighborhood relations by computerized texture analysis among pixels in these matrices and thus detect “hidden information” about texture changes that might not be visible for the human observer at the level of light microscopy.

## 2. Results

Unstained smears of peripheral blood from 45 patients with sickle cell disease (26 cases of SS, 12 cases of SC, 4 patients with S beta-thalassemia, and 3 cases of S alfa-thalassemia) were compared with those of 27 cases of persons without hematological disease. Median age of the patients was 43 years (17–72); there were 21 men and 24 women. Among the SCD patients, 20 were receiving hydroxyurea, including 18 of 26 with SS and 2 of 4 with S beta-thalassemia. 

We analyzed 10 cells per case in controls and 10 non-sickled cells per patient, but only a mean of 5 sickled cells per smear of patients, since this type was rarely found in some blood films. Therefore, a total of 270 cells from the control group and 450 non-sickled and 214 sickled cells of patients were examined.

The morphological characteristics of the images are shown in Figure 1 (normal RBCs) and patients (Figure 2, Figure 3 and Figure 4). 

In the normal control RBCs, we observed a homogeneous cytoplasm, which showed a deep blue color in the default FLIM picture. Interestingly, in SCD, one could detect 0–3 small dots (white/bluish in the FLIM images) that were never observed in normal red blood cells. They were also present in sickled cells. These cells also revealed some irregularly shaped large areas adherent to the cell membrane, also intensively fluorescent with increased lifetime values. 

Counting the small dots and larger areas together, the total foci per cell increased from normal-looking cells to sickle-shaped RBCs in SCD patients (Figure 4) (mean 0.05/cell and 1.63/cell, respectively; *p* < 0.001). Considering only sickled cells, the number of dots per cell was highest in patients with Sβ thalassemia (*p* = 0.04). There were no significant differences in the number of dots per cell between patients treated with hydroxyurea and those not treated.

Table 1 shows FLIM matrix-derived variables expressed as mean values per patient. The variables “mean”, “standard deviation”, “skewness”, and “entropy” of the lifetime histograms were not significantly different when comparing SCD patients and controls but reached a high level of significance when derived from LBP transformation of these variables. No feature was significantly different between SS and SC or S thalassemia, and so they were calculated together (Figure 5).

Finally, we ran a stepwise linear discriminant analysis based on the four variables derived from LBP-transformed matrices. The final model contained the variables “LBP mean” and “LBP entropy”, with an accuracy of 84.7%. This result dropped to 83.3% after the leave-one-out procedure.

In a further step, we analyzed the differences between single cells. Table 2 shows FLIM matrix-derived variables expressed as mean values per cell. Significant differences were found in “mean” and “entropy”, and in all variables after the LBP transformation (Figure 6). In the stepwise linear discriminant analysis, a final model was found based on the variables “LBP mean” and “LBP entropy” that yielded an accuracy of 68.5%, which dropped 68.4% after the leave-one-out procedure. The machine learning algorithm was performed according to the linear discriminant analysis classifier (LDA). In this way, we achieved an average F-score of 0.68 ± 0.01. The average accuracy was 68.1%.

## 3. Discussion

In the present study, we aimed to examine whether the FLIM technique, used in label-free routine peripheral blood smears, was suitable for the detection of the process of polymerization of hemoglobin S in SCD patients. Hemoglobin is considered weakly fluorescent, but an auto-fluorescence has been repeatedly shown in cytological or histological preparations [10,18,19,20]. This paradox can be explained by the observation that immediately after an exposure to a light beam, hemoglobin is transformed into a fluorescent photoproduct [18]. These results have been confirmed by Radmilovic et al. [6], who showed that the photoproducts were stable for several weeks. Furthermore, these authors demonstrated that several molecular species of photoproducts had been formed by the ultra-short laser stimuli. Interestingly, these were nearly identical to those of a previous experiment by Nagababu et al. [21], where hemoglobin had been denatured by non-neutralized hydrogen peroxide, creating very similar auto-fluorescent degradation molecules. Therefore, it can be hypothesized that during the pathogenesis of the laser-induced photo-degradation process, hydrogen peroxide may be initially involved. 

Air-drying is considered a fast preparatory procedure for hematological smears, interrupting cell metabolism in a few seconds. This technique has been used for more than one hundred years in a myriad of laboratories. Storage of the air-dried smears is only possible for a short time since, after a few days, degradation processes are so intense that staining is impaired. Fixation with formaldehyde vapor has shown not to change cell morphology or affect cell membrane interactions [22,23,24]. 

For research in sickle cell disease, it is important to have a reliable imaging technique for single-cell analysis. In particular, this is important for the evaluation of the spatial distribution of hemoglobin and its polymerization products. Analyses of routinely stained blood smears of sickled RBCs do not clearly reveal areas of polymerization [25]. Transmission and scanning electron microscopy and atomic force microscopy are considered excellent options for the study of ultrastructural details of erythrocytes. However, these methods usually need the application of immersion fixation and labeling, which might modify the cell structures. Moreover, these techniques are rather time-consuming, and therefore only a small number of cells can be examined. Several microscopic techniques have been created for unstained biological material, which are based on the different optical characteristics between the polymerized fibers and the surrounding cytoplasm, such as phase contrast, optical birefringence microscopy, dynamic and static light scattering, and differential interference contrast microscopy [3,4,5,6,7,8,9,26]. Their advantage is the fast detection of structural changes in unlabeled material, but they do not provide sufficient details for morphological analysis. These techniques highlight differences in the optical behavior of cellular structures, which are largely independent of their chemical composition. In this context, techniques based on the visualization and analysis of auto-fluorescence light are interesting, and because they do not use staining, give the option of confocal microscopy, which provides higher image quality, and finally, permit us, via the emission spectra, to get some information about the chemical composition of the material. A more comprehensive approach to this last topic can be achieved by the FLIM technique [11,12,13,14,15,16,17]. In this context, a lifetime is equivalent to the time delay of the photon emission after the absorption of the laser pulse by the molecules. The lifetime depends not only on the chemical composition of the cell but also on its microenvironment [12,13,14,15,16,17], but is independent of the concentration of the fluorophores. During polymer growth, the lifetimes increase. This has already been shown for some molecules [11], and we could confirm this phenomenon for HbS, which, to our knowledge, has not been reported by other groups so far. In that way, FLIM shows not only good-quality images due to the use of confocal microscopy but also gives additional information not revealed by other microscopic techniques. Moreover, the method is user-friendly, robust, and reproducible [10,11,12,13,18,19,20].

In our material, we found dot-like structures with a high FLIM value that were present only in RBCs from patients with SCD. These intensively fluorescent dots were never found in erythrocytes of controls or outside of RBCs. Furthermore, our images are very similar to those obtained by experiments with other microscopic technologies. Therefore, we think that our images show real structures and not artefacts. The same was true for the irregularly shaped, large, highly fluorescent areas in sickled cells. The increased lifetimes suggest changes in the molecular structure. It is well known that polymerization of a molecule is associated with an increase of its lifetime values [10]. This also favors the hypothesis that the prominent areas in the FLIM pictures represent polymerized HbS. 

Clusters are believed to be transient protein oligomers, probably due to partial protein unfolding and hydrophobic stabilization followed by oligomerization and cluster formation. It has been hypothesized that clusters have an inhomogeneous structure with an inside core of high viscosity. Partial protein unfolding and hydrophobic stabilization are considered the principal factors for oligomerization and cluster formation. The initiation of HbS polymerization is described by the double-nucleation model, which states that the nuclei can start from HbS monomers or on the surface of pre-existing fibers. Cluster sizes are comparable to the diffraction limit. A single fiber is composed of 14 chains of HbS tetramers with a seven-double-strand configuration. Fibers may tightly bind together in parallel and grow as a bundle. The double nucleation model describes that nuclei can form either from HbS monomers (homogeneous nucleation) or on the surfaces of pre-existing HbS (heterogeneous nucleation). In the latter situation, branched structures will appear. The final cell shape is a result of the intracellularly aligned hemoglobin polymers and the effective sickle hemoglobin polymer growth rate. Free hematin, spontaneously released by HbS due to its intrinsic instability, intensively accelerates nucleation and growth of the polymers [3].

Reoxygenation can reverse sickle cell deformation and occurs in vivo. Experimental studies have shown that, once having undergone a sickling event, the erythrocyte always deforms along the same axis during subsequent cycles and retains a ‘memory’ of its previous sickling cycles [7,8].

Using texture analysis and machine learning applied to the matrices derived from the FLIM data and comparing patients and controls, no differences could be found for the lifetime variables mean, standard deviation, skewness, or entropy of round, normally shaped RBCs from patients and controls. But after the LBP transformation of the matrices, significant differences between control RBCs and morphologically normal RBCs of SCD patients could be revealed. Finally, the linear discriminant analysis was able to correctly classify about 84% of the cases of patients and controls. When comparing the FLIM images of the control RBCs and the normally shaped non-sickled RBCs of the patients, a human observer can only see in the latter some highly fluorescent dots, which are only present in a small number in some of the patients’ RBCs. These small dots occupy an insignificant part of the cell volume and certainly cannot explain the significant differences found in the texture analysis. These observations suggest that subtle changes in the structure of the hemoglobin matrix before major polymerization are not seen by light microscopy but are detectable by computerized texture analysis. In that way, the presented FLIM technique with subsequent texture analysis opens new ways of investigation that cannot be obtained by the microscopic techniques mentioned above. Especially in the case of SCD research, the proposed technique could be useful for the study of new drugs, in particular, when they target the inhibition of the polymerization process.

Our investigation has some limitations. In order to provoke sickling of some cells in the blood samples collected outside a vaso-occlusive crisis, we kept the blood for 24 h at 4 to 8 °C, which is different from the in vivo process. Since the fluorescence of hemoglobin in erythrocytes is based on the formation of a hemoglobin photoproduct during irradiation [21,27], it would be necessary to investigate the biochemical nature of the fluorescent polymerized hemoglobin in our study. Due to technical reasons, it was only possible to extract matrix data inside of a rectangular region of interest, which, of course, is smaller than the whole area of a RBC. Nevertheless, we could reveal differences in the matrix structures between different cells from controls and patients. Due to the relatively long exposure time of eight minutes per frame, the number of acquired cells per patient was relatively small. It would be desirable to examine large numbers of non-sickled RBCs in SCD patients in order to see whether there is a transition gradient from the normal cell to the state with dot formation within single patients. Furthermore, the limited number of 45 patients is rather small when we are looking for differences in disease subgroups. Finally, it would be desirable to examine fresh blood, especially during a sickle cell crisis, and also simultaneously compare, in new investigations, the FLIM technique together with the above-mentioned techniques.

Even so, our study suggests that auto-fluorescence and fluorescence lifetime imaging of unstained blood films of patients with SCD are able to reveal the topography of the polymerization in RBCs and indicate differences in the molecular structure of the intracellular hemoglobin arrangement before nucleation. The highly fluorescent dots disclosed in the FLIM image might show a nucleation of the polymerization process. So, our study demonstrates the topography of the intracellular polymerization of hemoglobin in sickle cell disease.

Additionally, the textural analysis of the FLIM images reveals changes in hemoglobin arrangement not visible by light microscopy in normal-shaped RBCs of patients with SCD, and several texture features have high accuracy in distinguishing normal RBCs from those with hemoglobin S. Therefore, we think that the presented technique may be an interesting tool for the investigation of therapeutic inhibition of HbS polymerization. From a general point of view, our investigation demonstrates that computerized texture analysis of a FLIM matrix may reveal very important additional quantitative information when compared to the qualitative analysis by a human observer. 

## 4. Material and Methods

### 4.1. Patients

Our study was approved by the local ethics committee (proc. 50809815.7.0000.5404). Air-dried smears from peripheral blood collected in routine control appointments of SCD patients were examined. After routine peripheral blood counts were made, the sample was stored for 12–24 h at a temperature of 4–8 °C before a smear was performed. The air-dried slides were fixed by formaldehyde vapor for 1 h and stored at room temperature [18,19,20,27]. Normal RBCs from patients without a hematological disease were used as the control group.

### 4.2. Techniques

Analysis was made as formerly described [10]. Slides were irradiated by a pulsed-diode laser (405 nm, 80 MHz). Images were acquired by a confocal Zeiss Upright LSM780-NLO microscope equipped with a 63x oil immersion objective with a ×3 electronic zoom and an HPM-100-40 Hybrid detector (Becker & Hickl, Berlin, Germany). Photons were captured by a photon-counting PMT detector (Becker & Hickl GmbH, Berlin, Germany) and analyzed by a time-correlated single-photon-counting capture card (Becker & Hickl, SPC-830, Berlin, Germany), applying the software SPCM-Single Photon Counter 9.76 and SPCImage 6.2). For each image of 512 × 512 pixels, photons were collected for 8 min (according to the recommendation of the manufacturer) in order to get enough photons per pixel, and the mean fluorescence lifetime was calculated for each pixel. In order to reduce noise, we applied the bin-1 function. Then, a virtual FLIM image was constructed by attributing artificial colors to these mean lifetime values (Figure 1 and Figure 2), similar to the electromagnetic spectrum of visible light, starting with blue for the shortest and ending with red for the longest lifetime. At each measurement session, five RBCs from a formaldehyde-fixed reference slide were also measured.

For each RBC examined, we set a rectangle limiting the maximum possible area of interest inside the RBC, strictly avoiding the inclusion of extracellular areas containing plasma. The rectangle defined a matrix containing the mean values per pixel. From these matrices, we calculated the mean fluorescence lifetime per cell, its standard deviation, skewness, and entropy. In a further step, this matrix was LBP-transformed. The new matrix yielded a vector, later used for machine learning. Furthermore, we also calculated the mean, its standard deviation, skewness, and entropy for linear discriminant analysis. 

### 4.3. Local Binary Patterns

Local binary pattern (LBP) is a well-established technique to describe digital images, with a special focus on texture images and face recognition [28]. Its straightforward definition and interpretation, as well as the good results achieved in practice, made it a popular tool in classical computer vision (before the popularization of end-to-end deep learning approaches) [29]. Among the reasons for that success, one may mention its low computational cost and ability to capture spatial patterns, preserving robustness to illumination changes.

The essential idea of LBP is to compare each pixel in the image with its neighbors within a pre-defined radius, assigning 1 if the neighboring pixel has greater intensity than the central pixel and zero otherwise. The LBP code corresponds to that binary representation, preserving the order of the neighboring pixels. For practical purposes, that sequence of binary values is converted to decimal representation. Mathematically, in a specific position (*x*,*y*) of the image we have
(1)$$LBP_{P,R}\left(x,y\right)=\sum_{i=0}^{P-1}s(g_{i}-g_{c})2^{i}$$
where *R* is the neighborhood radius, *P* is the number of neighboring pixels, *g_c_* is the intensity of the central pixel, *g_i_* is the intensity of the *i*-th neighboring pixel, and *s(x)* is an indicator function, such that *s(x)* = 1 if *x* ≥ 0, otherwise *s(x)* = 0. Finally, the image descriptors are provided by the histogram of LBP codes of the image.

Here, in particular, we employ a rotation-invariant uniform version of LBP [30]. To enforce rotation-invariance, the minimum LBP code taken under circular bitwise shifts of the neighborhood is considered. As for the uniformity, a binary pattern is uniform if it contains, at most, two 0–1 or 1–0 transitions. The uniform version of LBP treats uniform patterns as a unique code in the computation of the histogram. Mathematically, it can be defined by
(2)$$LBP_{P,R}^{riu2}=\left\{\begin{array}{c} \sum_{i=0}^{P-1}s\left(g_{p}-g_{c}\right)2^{i}\\ P+1 \end{array}\right.\begin{array}{ll} & if\;U\left(LBP_{P,R}\right)\geq 2\\ & otherwise, \end{array}$$
where *U* is the uniformity function:(3)$$U(LBP_{P,R})=|s(g_{P-1}-g_{c})-s(g_{0}-g_{c})|+\sum_{i=1}^{P-1}|s(g_{i}-g_{c})-s(g_{i-1}-g_{c})|$$

*P* and *R* are parameters of the method to be defined empirically. Here we adopt *P =* 24 and *R* = 3. The pixel intensities at any real coordinate over the circle with radius *R* are obtained by bilinear interpolation over the pixels effectively present on the original image. Such a procedure gives rise to 26 possible LBP codes, and the descriptors themselves (histogram) correspond to the number of each of these codes in the image. Thus, we have a total of 26 LBP descriptors for each image.

### 4.4. Linear Discriminant Classifier 

We also ran the same task with a machine learning algorithm. More specifically we employed the linear discriminant analysis classifier (LDA) [31]. Its primary objective is to find a linear combination of features that best separate the classes. For that, the method assumes that the data from different classes are normally distributed and share the same covariance matrix (but with different means). The goal is to maximize the separation between the means while minimizing the spread within each class. This is performed by finding a set of linear boundaries (hyperplanes) that separate the classes. It projects the data into a new lower-dimensional space where the classes are as distinct as possible.

Mathematically, it works by maximizing the Fisher criterion, defined as:(4)$$J\left(w\right)=\frac{w^{T}S_{B}w}{w^{T}S_{B}w.}$$
where w is the linear transformation vector, SB is the inter-class scatter matrix, and SW is the intra-class scatter matrix. LDA chooses a linear combination of features that maximizes this criterion, increasing, in this way, the class separability. We adopted a hold-out validation scheme (50%/50% for training/testing), and the procedure was repeated 10 times.

### 4.5. Statistical Analysis

Data were analyzed on two levels. First–cell level: the data represent each cell. Second–patient level: the data represent the mean level of the cells analyzed in that patient. The presence of normal distribution was tested according to Kolmogorov–Smirnov. When comparing sickled and non-sickled cells of SCD patients, we used tests for dependent groups, and for the comparison to controls, we used tests for independent groups. For group separation, we ran a stepwise discriminant analysis [32,33] followed by the leave-one-out method in order to avoid overfitting. For statistical analysis, Winstat 3.1 and SPSS 22 were used.

## Figures and Tables

**Figure 1 ijms-25-12290-f001:**
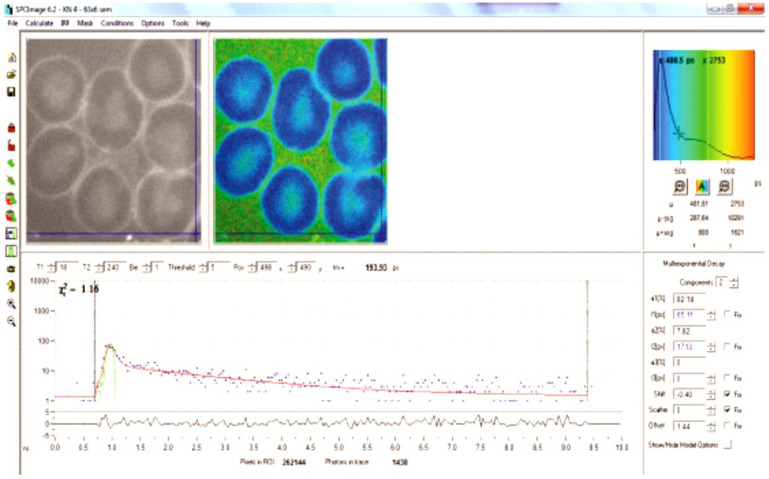
Image of a peripheral blood film of a control case with several normal RBCs. Upper left: auto-fluorescence picture. Upper middle: fluorescence lifetime image: the blue color corresponds to the lifetime of hemoglobin. Surrounding plasma in green/yellow color corresponding to a higher lifetime. A cursor is placed on a RBC (right inferior corner). Upper right: histogram of the lifetime distribution of the image (pseudo-colors according to the rainbow spectrum). Blue represents the shortest lifetime and red is the longest. The histogram shows that hemoglobin has a short lifetime. Below is the fluorescence lifetime decay curve of the selected spot in the image. Every dot represents a single photon.

**Figure 2 ijms-25-12290-f002:**
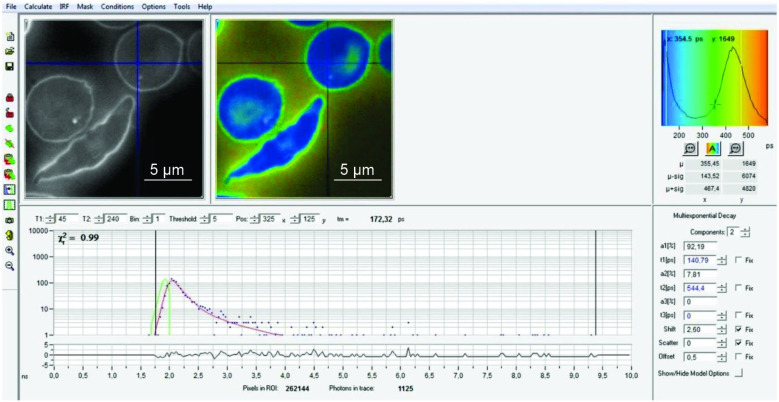
Upper left: autofluorescence and FLIM images of a patient with homozygous SS hemoglobinopathy. Two normal-shaped and one sickled RBC. Each of the normal looking ones shows one highly fluorescent dot. The sickled RBC has areas with a higher fluorescence suggestive of polymerization. The histogram on the right side represents the lifetime of the region where the cursor is placed. Lower right is the decay curve of the selected region of interest.

**Figure 3 ijms-25-12290-f003:**
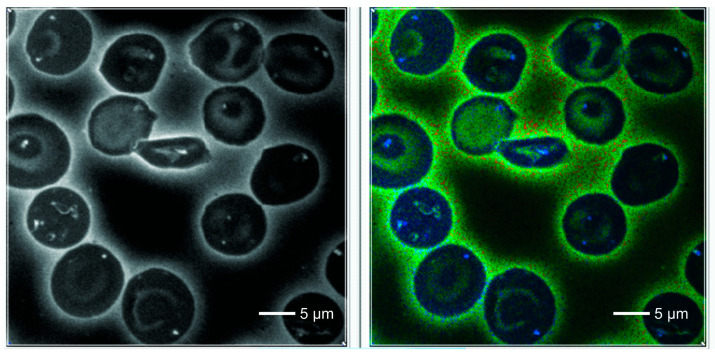
RBCs from a blood smear of a patient with SC hemoglobinopathy. Left: auto-fluorescence and right FLIM image. In the center, a sickled cell with an irregular heterogeneous area of enhanced fluorescence revealing a higher lifetime value in the FLIM compared to the surrounding cytoplasm. Some of the non-sickled RBCs show highly fluorescent dots.

**Figure 4 ijms-25-12290-f004:**
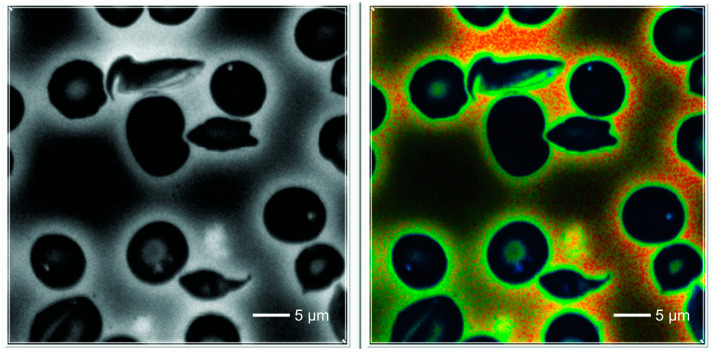
RBCs from a blood smear of S beta-thalassemia hemoglobinopathy. Three entire sickled cells with irregular, sometimes heterogeneous areas with enhanced fluorescence revealing higher lifetime values compared to the surrounding cytoplasm. Some of the non-sickled RBCs show highly fluorescent dots.

**Figure 5 ijms-25-12290-f005:**
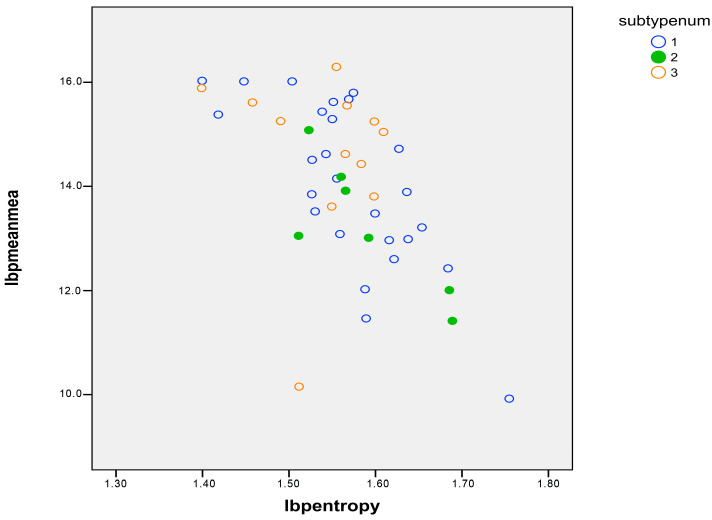
Distribution of non-sickled cells in patients with SCD according to the sub-types: SS in black, SC in green, and S thalassemia in orange. There were no significant differences among the different sub-types of SCD.

**Figure 6 ijms-25-12290-f006:**
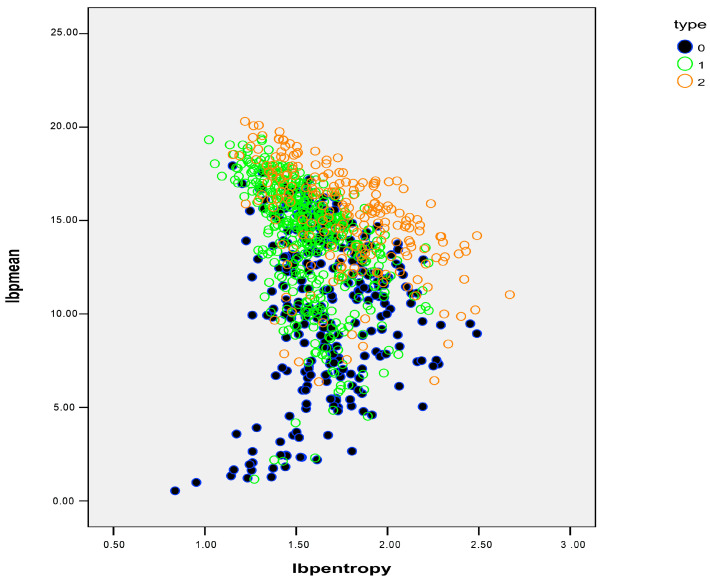
Dot plot showing the distribution of LBP mean (Y axis) and LBP entropy (X axis) to show the distribution of each cell in the control group (blue), and SCD: non-sickled cells are in green and sickled cells are in orange. Both parameters were able to discriminate between normal and SCD in 84.7% of the cases.

**Table 1 ijms-25-12290-t001:** Mean FLIM values in picoseconds per case of red blood cells in controls and patients.

	Control Cases N = 27	*p*-Values * Control × not Sickled	Non-Sickled Cells from Patients N = 45	Sickled Cells from Patients N = 34	*p*-Values ** Non-Sickled × Sickled RBCs
Mean	175.689	0.31	192.343	247.568	0.000
St. deviation of Mean	30.050	0.61	27.702	56.240	0.000
Entropy	1.883	0.19	1.8597	1.7884	0.033
Skewness	0.801	0.18	0.637	1.033	0.005
LBP					
Mean	10.683	0.000	14.063	15.222	0.003
St. deviation of Mean	10.284	0.001	10.959	10.498	0.000
Entropy	1.675	0.000	1.564	1.721	0.000
Skewness	0.649	0.000	−0.1241	−0.300	0.005

* Mann–Whitney test: ** Wilcoxon test.

**Table 2 ijms-25-12290-t002:** Mean FLIM values in picoseconds per individual red blood cell in controls and patients.

	Controls N = 270	*p*-Values * Control × not Sickled	Patients, not Sickled N = 432	Patients, Sickled N = 214	*p*-Values * Non-Sickled × Sickled RBCs
Mean	175.531	0.004	193.891	253.335	0.000
St. deviation of mean	30.222	0.49	28.091	58.614	0.000
5%	135.700	0.000	155.050	181.516	0.000
95%	227.679	0.068	243.867	361.172	0.000
Entropy	1.886	0.019	1.866	1.807	0.010
Skewness	0.787	0.072	0.630	0.950	0.000
LBP					
Mean	10.683	0.000	13.981	15.097	0.000
St. deviation of mean	10.284	0.001	10.945	10.475	0.000
Entropy	1.675	0.000	1.567	1.766	0.002
Skewness	0.649	0.000	−0.103	−0.317	0.000

* Mann–Whitney test.

## Data Availability

Patients’ data are subjected to privacy restrictions but, if necessary, the authors can share the data.

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
