# Peer review of "Accompanying Hemoglobin Polymerization in Red Blood Cells in Patients with Sickle Cell Disease Using Fluorescence Lifetime Imaging"

_ijms, 2024, doi:10.3390/ijms252212290_

Round 1
Reviewer 1 Report
Comments and Suggestions for Authors
Da Silva et al. reported research findings on FLIM imaging of sickle cell disease. The study could have important applications in diagnosing and monitoring disease treatment.
Before publishing, certain flaws need to be addressed.
1. Introduction:
a) Clarify the Research Objectives: The text could benefit from a more explicit statement of the research objectives. Clearly outlining the aims and hypotheses of the investigation would help provide a stronger foundation for the study.
b) Please add paragraphs explaining the comparison of FLIM with other microscopies used for SCD RBC analysis, emphasizing prons and cons
2. Results:
a) The main issue I found is about sample preparation. The authors stated that the samples are air-dried samples; however, in the M&M section, it is said that samples are fixed in the formaldehyde vapor. It is known that fixation causes artifacts and morphology change. Additionally, air drying induces inevitable oxidation. So, all the above-mentioned factors reduce the usability of the presented results.
b) Please provide the results of one or two fresh blood samples from patients with Sickle Cell Disease (SCD) using FLIM, as well as light microscopy and either confocal or epifluorescent microscopy. This will help distinguish your hypothesis regarding polymerized hemoglobin from the already available data on hemoglobin autofluorescence (photo-product), which is generated after oxidation, as reported in previous studies of Nagababu et al (2003) https://doi.org/10.1016/S0304-4165(02)00537-8 and even confirmed recent one by Radmilovic et al. (2023) https://doi.org/10.1016/j.ijbiomac.2023.125312
c) In all figures, the first letter of the legend sentence should be capitalized. It applies to all tables. Additionally, Table Legends should be given above the tables.
d) FLIM images and their interpretation could be presented with greater clarity. This could involve breaking down complex concepts into more digestible segments and ensuring that the implications of the findings are conveyed. Please also describe in more detail how your results support the two-step nucleation theory of hemoglobin polymerization.
e) Figures 1 and 2 should be presented in better resolution
3. The conclusion could be strengthened by emphasizing the practical implications of the study's findings. Exploring how the FLIM technique could impact the investigation of therapeutic inhibition of polymerization and its broader applications would enhance the conclusion.
Author Response
Comments and Suggestions for Authors
Da Silva et al. reported research findings on FLIM imaging of sickle cell disease. The study could have important applications in diagnosing and monitoring disease treatment.
We want to thank the reviewer for the valuable suggestions that let us improve our manuscript.
Changes made in the text are in red.
Before publishing, certain flaws need to be addressed.
- Introduction:
- a) Clarify the Research Objectives: The text could benefit from a more explicit statement of the research objectives. Clearly outlining the aims and hypotheses of the investigation would help provide a stronger foundation for the study.
We revised the last paragraph of the introduction in order to make our aims more clear.
- b) Please add paragraphs explaining the comparison of FLIM with other microscopies used for SCD RBC analysis, emphasizing prons and cons
we included this topic in the discussion explaining in more detail, the techniques that have been used for the study of polymerization of hemoglobin S, and cited the most important works in the references.
- Results:
- a) The main issue I found is about sample preparation. The authors stated that the samples are air-dried samples; however, in the M&M section, it is said that samples are fixed in the formaldehyde vapor. It is known that fixation causes artifacts and morphology change. Additionally, air drying induces inevitable oxidation. So, all the above-mentioned factors reduce the usability of the presented results.
We added 2 references explaining the role of air-drying in hematological smears, and the choice of formaldehyde vapor for fixation.
- b) Please provide the results of one or two fresh blood samples from patients with Sickle Cell Disease (SCD) using FLIM, as well as light microscopy and either confocal or epifluorescent microscopy. This will help distinguish your hypothesis regarding polymerized hemoglobin from the already available data on hemoglobin autofluorescence (photo-product), which is generated after oxidation, as reported in previous studies of Nagababu et al (2003) https://doi.org/10.1016/S0304-4165(02)00537-8 and even confirmed recent one by Radmilovic et al. (2023) https://doi.org/10.1016/j.ijbiomac.2023.125312
the time given for manuscript revision precluded making new experiments with suitable patients that would agree to participate in the study. Only the approval by the Ethics Commitee would last some months. This could be the subject of a next research.
- c) In all figures, the first letter of the legend sentence should be capitalized. It applies to all tables. Additionally, Table Legends should be given above the tables.
All this was corrected. It was a problem in the formatting of the text by MDPI
- d) FLIM images and their interpretation could be presented with greater clarity. This could involve breaking down complex concepts into more digestible segments and ensuring that the implications of the findings are conveyed. Please also describe in more detail how your results support the two-step nucleation theory of hemoglobin polymerization.
We extended the figure caption of figure 1 and 2 in order to explain better how the measures were made.
- e) Figures 1 and 2 should be presented in better resolution
figure 2 was substituted for a better one.
Concerning figure 1, we found no better figure. In label-free cytology it is difficult to obtain a better defined image in FLIM, but it has sufficient brightness and contrast to be compared with the other figures.
- The conclusion could be strengthened by emphasizing the practical implications of the study's findings. Exploring how the FLIM technique could impact the investigation of therapeutic inhibition of polymerization and its broader applications would enhance the conclusion.
Introduction and discussion were thoroughly revised.
Reviewer 2 Report
Comments and Suggestions for Authors
The paper by Borges da Silva et al. is quite interesting and potentially relevant.
Indeed it deserves publication but I hink that its relevance does not consist on what they claim (for the early detection of sickle cell patients) but for a better comprehension of the mechanism of intracellular polymerization and eventually for the prognosis of the disease in a given patient. Thus, a very important point is the intraerythrocytic presence of small hemoglobin polymers even under oxygenation conditions and how they are formed (see Coletta et al., 1988, FEBS Lett. 236, 127-131). As a matter of fact, authors should report on several red blood cells from homozygotes and heterozygotes sickle cell patients and determine the extent of intracellular polymerization present even in oxygenated conditions. I think that informations and statistics on the presence and the percentage of polymerized hemoglobin in oxygenated erythrocytes is a crucial information for predicting the pathological evolution of a given patient.
Author Response
Comments and Suggestions for Authors
The paper by Borges da Silva et al. is quite interesting and potentially relevant.
We thank the reviewer for the valuable suggestions and the positive comments.
Changes made in the text are in red text are in red.
Indeed it deserves publication but I think that its relevance does not consist on what they claim (for the early detection of sickle cell patients) but for a better comprehension of the mechanism of intracellular polymerization and eventually for the prognosis of the disease in a given patient. Thus, a very important point is the intraerythrocytic presence of small hemoglobin polymers even under oxygenation conditions and how they are formed (see Coletta et al., 1988, FEBS Lett. 236, 127-131). As a matter of fact, authors should report on several red blood cells from homozygotes and heterozygotes sickle cell patients and determine the extent of intracellular polymerization present even in oxygenated conditions. I think that informations and statistics on the presence and the percentage of polymerized hemoglobin in oxygenated erythrocytes is a crucial information for predicting the pathological evolution of a given patient.
We agree with the reviewer, but our study design was to describe the morphology and the texture analysis of peripheral blood in patients with SCD. Please see our revised description of our aims (revised introduction)
Indeed an additional study as suggested by the reviewer would be interesting, but for this purpose we would have to submit a new project to our Ethics Committee, and the reply would be given after 3 months. Please remind the ethical laws in Brazil are more restrictive than in many other countries. In summary, first results would only be obtained next year.
Round 2
Reviewer 2 Report
Comments and Suggestions for Authors
I understand the motivations for delaying the advised experiments to another publication.
However, I think that the experiments will be highly relevant and I hope they will carry them out quickly